# *Brucella abortus* Strain RB51 Administered to Prepubescent Water Buffaloes, from Vaccination to Lactation: Kinetics of Antibody Response and Vaccine Safety

**DOI:** 10.3390/microorganisms11082078

**Published:** 2023-08-13

**Authors:** Fabrizio De Massis, Flavio Sacchini, Nicola D’Alterio, Giacomo Migliorati, Nicola Ferri, Emanuela Rossi, Daniela Averaimo, Antonio Petrini, Michele Podaliri Vulpiani, Fabrizia Perletta, Diamante Rodomonti, Mirella Luciani, Giovanni Befacchia, Marta Maggetti, Tiziana Di Febo, Chiara Di Pancrazio, Ivanka Marinova Krasteva, Romolo Salini, Giacomo Vincifori, Simona Iannetti, Manuela Tittarelli

**Affiliations:** Istituto Zooprofilattico Sperimentale dell’Abruzzo e del Molise, 64100 Teramo, Italy; f.demassis@izs.it (F.D.M.); n.dalterio@izs.it (N.D.); g.migliorati@izs.it (G.M.); n.ferri@izs.it (N.F.); e.rossi@izs.it (E.R.); d.averaimo@izs.it (D.A.); a.petrini@izs.it (A.P.); m.podaliri@izs.it (M.P.V.); f.perletta@izs.it (F.P.); d.rodomonti@izs.it (D.R.); m.luciani@izs.it (M.L.); m.maggetti@izs.it (M.M.); t.difebo@izs.it (T.D.F.); c.dipancrazio@izs.it (C.D.P.); i.krasteva@izs.it (I.M.K.); r.salini@izs.it (R.S.); s.iannetti@izs.it (S.I.); m.tittarelli@izs.it (M.T.)

**Keywords:** brucellosis, water buffalo, *Brucella abortus* strain RB51, vaccine safety, vaccination, complement fixation test

## Abstract

*Brucella* RB51 is a live modified vaccine. Its use in water buffalo has been proposed using a vaccination protocol different to that used for cattle, but knowledge of the long-term effects of RB51 vaccination in this species remains incomplete. The aim of the study was to evaluate the safety and kinetics of antibody responses in water buffaloes vaccinated according to the protocol described for the bovine species in the WOAH Manual, modified with the use of a triple dose. Water buffaloes were vaccinated with the vaccine RB51. A booster vaccination was administered at 12 months of age. When turning 23–25 months old, female animals were induced to pregnancy. RB51-specific antibodies were detected and quantified using a CFT based on the RB51 antigen. Vaccinated animals showed a positive serological reaction following each vaccine injection, but titers and the duration of the antibody differed among animals. For 36 weeks after booster vaccination, the comparison of CFT values between vaccinated and control groups remained constantly significant. Afterwards, antibody titers decreased. No relevant changes in antibody response were recorded during pregnancy or lactation. In conclusion, results indicated that the vaccination schedule applied is safe and allows for vaccinated and unvaccinated controls to be discriminated between for up to 8 months after booster vaccination.

## 1. Introduction

Brucellosis is one of the most important zoonotic diseases worldwide and is responsible for heavy economic losses due to late term abortions, stillbirths, and parturition of weakly calves [1]. The disease is also a serious public health problem wherever the infectious agent is present.

Water buffalo, particularly in Mediterranean and Asian countries, represent an important livestock due to its high-quality meat and milk. Water buffalo milk is rich in nutrients, including proteins, vitamins, and minerals. It contains a higher fat content and different fatty acid profile compared to cow’s milk, making it a valuable source of nutrition for human consumption. It is also used to produce traditional dairy products such as cheese, butter, and Italian buffalo mozzarella. Dairy products derived from water buffalo have economic importance and contribute to human health and nutrition where this species is farmed [2].

In Italy, brucellosis in water buffaloes (*Bubalus bubalis*) still today represents a health priority, especially in some areas of the Campania region, where it is the highest. For this reason, in recent years, a series of special measures have been implemented in the attempt to control disease spread in Southern Italian territories including, in particular, the authorization for vaccination according to Ministerial Decree 651/94, with the possibility of using the *Brucella abortus* strain RB51 vaccine in water buffaloes.

The RB51 vaccine is a live attenuated, rough variant of the virulent strain *B. abortus* 2308 [3], resistant to Rifampicin, and lacking the side chain O of the LPS. It does not induce the production of antibodies detectable with classical (and official) serological tests such as the Rose Bengal test (RBT) or the complement fixation test (CFT) [4,5,6,7]. In addition, specific antibodies are detected by RB51-CFT performed with the homologous antigen [8]. Therefore, RB51-vaccinated cattle can be easily differentiated from naturally infected animals supporting the application of test-and-slaughter and vaccination policies simultaneously [9]. In cattle, vaccination with RB51 induces an immune response dominated by a Th1 profile where CD8^+^ cytotoxic T-cells, IFN-γ-producing CD4+ T-cells together with Th17 cell subsets represent the key cell components [10]. Being live attenuated, RB51 may lead to abortion and/or may be shed through milk, when administered to adult pregnant animals [11,12,13,14,15,16,17,18,19]. The strain may also survive in fresh or ripened cheeses [20]. The vaccine strain also has residual pathogenicity for humans [21].

The RB51 vaccine has been recorded to be safe and effective in preventing abortion and, therefore, to control the spreading of the infection in cattle (*Bos taurus*) [22]. However, when considering its application in water buffaloes, to date there are no guidelines or protocols recognized at an international level for the use of RB51 in this animal species, and the indications reported in the Manual of Diagnostic Tests and Vaccines of the World Organization for Animal Health (WOAH, founded as OIE) only refer to cattle [23]. Literature data on use of RB51 vaccine in water buffaloes are scant and controversial. Studies carried out in Trinidad administering single or double the dosage recommended for cattle (1.0–3.4 × 10^10^ UFC), with or without booster, failed to protect water buffaloes from *B. abortus* infection [24,25]. Other studies carried out in Italy suggested that vaccination of pre-pubescent animals using a dose triple the one recommended for cattle followed by a booster vaccination with the same dose after 30 days was able provide a protective immunity [26,27,28].

These data suggest that the use of RB51 in water buffaloes requires a different vaccination protocol but also strict compliance with the vaccination schedule. This in order to reduce the risks for both human and animal health due to the possible excretion in milk and induction of abortion in pregnant females [12,13,14,19]. 

Side effects such as vaccine shedding and abortion are known to occur when the vaccine is administered to pregnant or lactating animals [13,14,19]. Thus, vaccination is recommended at calf hood age but the dosage and vaccination schedule to be applied remain a debated issue with very few and conflicting data available for guidance. 

As matter of fact, there are a lack of data on the long-term effects of the RB51 vaccine administered as a triple dose to pre-pubescent water buffaloes in terms of immune response (humoral) and possible consequences for critical periods in the epidemiology of brucellosis, namely pregnancy, calving, and lactation. 

If previous vaccination campaigns in the buffalo species of the Campania territory may have helped to reduce the prevalence of brucellosis, on the other hand, situations of incorrect application of vaccination or its use outside official controls have emerged in this area. The vaccination of adult animals (even pregnant or lactating ones) has been suspected, with possible risks of RB51 shedding in the environment and in the food chain together with difficulties for management of RB51 eliminating animals.

Therefore, the illegal use of the RB51 vaccine represents a damage to the *Brucella*-free accreditation status already obtained or to be obtained by farms and territories, where it is applied, and constitutes an important public health issue.

This situation made evident some gaps that persist for a thorough understanding of the use of the RB51 vaccine in water buffalo, and which require insights that are more scientifically based. In particular, on some important aspects such as:

The identification of an official vaccination scheme for which the physiological immune response and its duration (from puberty to lactation) are identified and measured;The identification of diagnostic criteria and protocols to identify any use of the RB51 vaccine outside the prescriptions given by the Competent Authority.

As regards the development of diagnostic tools able to identify animals vaccinated with RB51, in recent years a protocol has been developed for cattle, based on the use of the specific complement fixation test for RB51 (RB51-CFT) and a test of intradermal reaction with Brucellin antigen, obtained from the homologous vaccine strain [29]. This diagnostic protocol was also found to be effective in the buffalo species [30] when using the commercial Brucellin (Brucellergene OCB^®^ Synbiotics Corporation, Lyon, France), instead of homologous Brucellin. 

The aim of the study was to characterize in the long-term the physiological immune response induced by RB51 vaccination in water buffaloes, adapting the vaccination schedule for cattle as reported in the WOAH Manual of Diagnostic Tests and Vaccines for Terrestrial Animals [23], modified with the use of a triple dose. Moreover, the study aimed to assess the possible spreading of the vaccine strain from vaccinated to unvaccinated animals, under different physiological conditions. The onset and persistence of RB51-specific antibodies following vaccination, possible microbiological risks related to vaccine spreading, and the performance of RB51-CFT on a buffalo population not vaccinated with RB51 and officially free from brucellosis were also investigated.

## 2. Materials and Methods

### 2.1. Animals, RB51 Vaccination and Insemination Protocol

The animal study was approved by the Italian Ministry of Health (authorization n. 498/2016-PR). Thirty-six water buffaloes (34 females and 2 males) aged between 5 and 9 months were recruited from officially brucellosis-free farms, not participating in the vaccination program, and identified in the province of Salerno, Campania, Italy. Before selection, animals were screened serologically for brucellosis infection (official tests) or vaccination (RB51-CFT), *Chlamidia psittaci*, Para Influenza 3, Bovine Viral Diarrhea, Q fever (*Coxiella burnetii*), and *Yersinia enterocolitica* O:9. Animals testing negative to the serological screening were enrolled in the study.

Selected water buffaloes were moved to the trial site, where duly authorized premises were located in accordance with Italian and European laws regarding animal experimentation and animal welfare. All water buffaloes were individually identified using ear tags and electronic ruminal bolus and were assigned a short ID number from 1 to 36. The animals were housed in standard farming conditions, with ad libitum feeding and drinking water. Animals were monitored on a daily basis and therapeutic treatment delivered when required. The duration of the experiment was expressed as weeks post vaccination (wpv). After two weeks of acclimatization in the experimental stable, the day of vaccination (wpv0), 30 water buffaloes were injected with 6 mL of the commercial vaccine RB51, kindly provided by CZ Vaccines (RB-51 CZV^®^, Porriño, Pontevedra, Spain), which was injected subcutaneously in the caudal portion of the neck, on the right side. The vaccine, according to the manufacturer’s instruction, contained 10–34 × 10^9^ CFU/2ml. The amount of vaccine injected (6 mL) was a dose triple that indicated for cattle (10–34 × 10^9^ CFU/2mL) [26,28]. Six water buffaloes were included as controls (2 males and 4 females), inoculated with a placebo (sterile saline solution), and immediately mixed with vaccinated animals. Syringes containing the vaccine formulation of the placebo were numbered and veterinarians, animal care personnel, and laboratory technicians involved in the experiment were kept blinded about the vaccination status of the animals. The four females included in the control group were selected using the randomization function of Microsoft Excel^®^, version 2013 (Microsoft, Redmond, WA, USA). The two male water buffaloes were not injected. A booster vaccination was administered after the animals turned the age of 1 year (12–14 months), corresponding to 20 weeks post-vaccination (wpv20), in agreement with the administration schedule of RB51 vaccine reported for cattle in the Manual of Diagnostic Tests and Vaccines for Terrestrial Animals [23]. A summary of the experimental design is shown in Figure 1.

At the age of 23–25 months (wpv63–wpv71), female animals (both vaccinated and controls) were subjected to estrus synchronization and became pregnant combining artificial insemination (AI) and natural breeding, the latter performed by the two males belonging to the unvaccinated control animals. To guarantee the necessary supervision and ensure more animal welfare during delivery and perinatal days, animals were divided in two groups and estrus synchronization was carried out in order to have at least one-month distance between the deliveries of one group and the following. Estrus synchronization was induced by applying an intravaginal progesterone insert (CIDR—Controlled Internal Drug Release) together with intramuscular (i.m.) injection of 2 mL of GNRH (Gonavet Veyx^®^, Elanco Italia, Sesto Fiorentino, Florence, Italy). After 7 days, CIDRs were removed, and animals were injected with 5 mL of pregnant mare serum gonadotropin 5000 (PMSG) and 2 mL prostaglandin F2 alpha (PGF2α). After 48 h, 2 mL of GNRH were administered i.m. and the following day the AI was performed together with a final injection of 2 mL of GNRH. In addition to AI, animals were left in contact with the two males, to facilitate the natural breeding in case animals returned to estrus. Pregnancy was diagnosed in the early stages (30 and 60 days after AI) using trans-rectal echography, while clinical monitoring in later stages continued by rectal palpation.

The entire study was designed as a long-term trial and animals were monitored for 33 months after vaccination, covering the vaccination stage of prepubescent animals, to pregnancy, to delivery and lactation. In this article, the experimental timing referring to vaccination until the end of pregnancies (Phase I) was expressed as weeks post vaccination (from weeks 0 to 109 i.e., 2.5 years—post-vaccination). After delivery, data from individual animals were all aligned to the date of calving and timing was expressed as weeks post-delivery (wpd) (Phase II). Bacteriological data include samples collected at calving and during lactation (Phase II).

### 2.2. Sampling Protocol

Blood samples were collected from the jugular vein using vacuum tubes. Sample collection was carried out all along the trial, including vaccination, pregnancy, and lactation steps. To monitor the antibody response after vaccination, serum samples were collected weekly (wpv0–wpv46), fortnightly (wpv46–wpv67), and then monthly (wpv67–wpv109), depending on the phase of the study and covering from vaccination to calving. After calving, serum samples were collected weekly for 4 weeks and fortnightly for the following 6 weeks. After collection, blood samples were centrifuged at 1000× *g* for 15 min and separated sera were stored at −20 °C until testing. At calving, tissue samples of placenta and colostrum were collected for bacteriological examination. Vaginal swabs were performed every day for the first 7 days post-delivering, and then samples were collected twice a week for additional 3 weeks. From the day of delivery, milk samples were collected every day for the first 7 days, twice a week from weeks 2 to 4 post-partum, and once a week from months 2 to 6 (when milk was available).

### 2.3. Field Sampling (Serum)

To generate additional data on the specificity of RB51-CFT applied to water buffaloes in field conditions, a panel of 393 sera was collected from seven officially brucellosis-free buffalo farms located in the Regions of Piemonte and Lazio. An aliquot of serum for these samples was collected during the sampling activities carried out within the national eradication plan for brucellosis. Animals with ages above 12 months were included.

### 2.4. Laboratory Analysis

#### 2.4.1. Official Serological Tests for Brucellosis

Serum samples collected during the trial were tested with official serological methods for brucellosis (Rose Bengal test (RBT) and complement fixation test (CFT)), to verify that animals vaccinated with the rough strain RB51 do not produce antibodies typically detected against smooth *Brucella* after infection. The official tests were performed according the WOAH Manual of Diagnostic Tests and Vaccines for terrestrial Animals [23], and samples were tested using two-fold dilutions starting from 1:4.

#### 2.4.2. Serological Tests for RB51—Specific Antibodies

RB51-specific antibodies (IgM and IgG) were detected and quantified using a specific CFT based on an RB51 homologous antigen (RB51-CFT), as previously described [8,25,30]. Samples were analyzed using two-fold dilutions, starting from 1:4. Sera with an antibody titer equal or above (++++) 1:4 (cut-off) were considered as positive. This RB51-CFT was able to detect antibodies of vaccinated animals that result negative to conventional tests for brucellosis, which use *Brucella* antigens in the smooth phase.

#### 2.4.3. Post-Mortem and Bacteriological Analyses

Considering that RB51 is a live attenuated vaccine and animals were immunized with a triple dose, bacteriological analyses were carried out to detect strain persistence in tissue organs and to assess potential shedding of the vaccine strain. Both culture and molecular methods were applied on different matrices during the study period. 

Water buffaloes deceased during the vaccine trial were subjected to post-mortem examination and laboratory investigations to identify the cause of death, and to investigate the possible persistence of the vaccine strain in the body. Tissue samples were collected from lymph nodes (retropharyngeal, submandibular, prescapular, precrural, supramammary, iliac, meseraic, lymph nodes of the broad ligament of the uterus), spleen, liver, uterus (horn and body), kidneys, and udder. In addition, cotyledons, vaginal swabs and milk samples were collected and tested from live animals after calving.

At the end of the trial, animals were slaughtered and tissues samples were collected as described above. All samples were subjected to *Brucella* spp. search by microbiological isolation and molecular techniques, following the protocol in use at the National Brucellosis Reference Center (CRNB-IZSAM, Teramo, Italy). WOAH procedures for *Brucella* isolation were performed [23], including weekly subcultures from enrichment liquid Farrell onto solid media (Farrell and CITA). Molecular investigations to detect *B. abortus* spp. were carried out using a PCR according to Di Giannatale et al. [31].

### 2.5. Statistical Analysis

To take into account the uncertainty of the proportion of positive laboratory results over the total tests performed, a beta distribution was used to define the 95% confidence interval of the proportion accuracy. The uncertainty interval was defined as the difference between upper and lower 95% confidence limits. The 95% lower and upper credibility levels (L.C.I. and U.C.L., respectively, composing the Credibility Interval, C.I.) of the distribution frequency of positive results were calculated using a Bayesian approach with a beta distribution (n + 1; n − s + 1), where n is the total number of tested samples and s is the tested positive samples [32]. 

Statistical comparisons were performed on RB51-CFT results between vaccinated and control groups for each sampling moment using Mann Whitney’s non-parametric test. Values of *p* ≤ 0.05 were considered as significant.

Further linear and geometric mean regression analyses were applied to the RB51-CFT results of the vaccinated animals to evaluate the descending trend of the antibody kinetics and time limit beyond which the application of the RB51-CFT alone can evolve in diagnostic pictures that are difficult to interpret.

## 3. Results

### 3.1. Animals, Vaccination Protocol, and Insemination Protocol

After the first and booster vaccinations, none of the animals showed side effects at the site of injection or at a systemic level. The synchronization and insemination procedures resulted in 29 pregnant females out of 30. All pregnant females had normal parturition.

Four deaths occurred during the study, independently of vaccination: animal # 13 (wpv22), vaccinated; animal # 5 (wpv26), vaccinated; animal # 22 (wpv35), control; animal # 26 (wpv55), vaccinated. The post-mortem examination and related laboratory tests of the four water buffaloes that died during the trial did not show any correlation between the cause of death of the subjects and the trial in progress. The experiment then ended with 27 animals vaccinated and 5 controls.

### 3.2. Laboratory Analysis

#### 3.2.1. Official Serological Tests for Brucellosis

None of the vaccinated animals reacted positive to official RBT or CFT after RB51 first or booster vaccination.

#### 3.2.2. Serological Tests for Brucellosis and Kinetic of RB51 Specific Antibodies Measured by RB51-CFT

Following first vaccination, all animals showed seroconversion starting from 1-week post vaccination (wpv1) with the mean group of antibody titer peaking on wpv3, even if titers and the persistence of RB51 specific antibodies differed among animals (Figure 2a,b). Actually, between the first and booster vaccinations, 25 out of 30 (83%) vaccinated animals showed a transient increase in antibodies followed by a period of negativity that ranged from 2 to 17 weeks. Within two weeks after the booster vaccination, all animals showed a new increase of antibody titer with the mean value of the group peaking 3 weeks after the booster (wpv23). Afterward, antibody titers showed a progressive but not linear decrease and from wpv32 onwards, animals started testing seronegative but alternating with periods of negativity and periods of positivity with RB51-CFT values near the cut off (1:4 serum dilution). Following the booster vaccination, only 1 out of 30 vaccinated animals remained positive to RB51-CFT throughout the entire period of observation (wpv21–wpv109). 

We also investigated variations in RB51-CFT sensitivity after vaccination, in terms of the percentage of vaccinated animals correctly identified as vaccinated, and thus with an antibody titer ≥1:4. One week after first vaccination, 96.7% (29/30; CI: 99.2–83.3%) of vaccinated animals tested were correctly identified as vaccinated by RB51-CFT. One animal seroconverted later on wpv4. The percentage of animals correctly identified as vaccinated by RB51-CFT decreased progressively, starting from wpv14 with the minimum value of 16.7% (5/30; CI: 33.7–7.5%) recorded on wpv19. Two weeks after the booster vaccination (wpv22), RB51-CFT detected 100.0% (wpv21; CI: 100.0–90.8%) of vaccinated animals, and this result was continuously confirmed for 10 weeks (from wpv21 to wpv31). Afterward, the percentage of animals correctly identified as vaccinated decreased as a non-linear and fluctuating trend, with values ranging from 96.4% (wpv32, 27 animals out of 28; CI: 99.2–82.2%) to 7.4% (wpv71, 2 animals out of 27; CI: 23.5–2.3%) and with all vaccinated animals testing negative only on wpv109 (Figure 3). 

Interestingly, the simultaneous negativity of all vaccinated animals was never registered during the sampling period considered, and no relevant increase or decrease in antibody titers was observed during pregnancy or after delivery (Figure 2a and Figure 4a). Moreover, after calving, the percentage of animals testing positive to RB51-CFT ranged between 16% (CI: 34.9–6.6%) and 3.8% (CI: 19.0–0.9%) (Figure 4b).

None of the controls (4 females and the two males involved in mating activities) showed any serological positivity in the period considered, including the pregnancy and lactation phases. This accounts for a specificity of 100% (CI 65.2–100% at the beginning of the experiment, 60.7–100% at the end of the experiment) (Appendix A).

Statistical analyses carried out using the Mann–Whitney test showed a significant difference in antibody titers between vaccinated and controls (*p* < 0.05) starting from wpv1 to wpv15 and on wpv18. After the booster vaccination, the serological response was significantly higher in vaccinated animals from wpv20 to wpv56, and from wpv61 to wpv71 with the last significant recorded on wpv84 (Figure 2a, Appendix A).

RB51-CFT analyses were also carried out in a field population of water buffaloes from officially Brucella-free farms to verify test specificity. Serum samples (393) were tested and only one animal resulted positive at the cut-off value (1:4). 

#### 3.2.3. Bacteriology

All samples tested negative to both culture and PCR, and a summary of matrices collected during different stages of the trial is displayed in Table 1. The analyses performed on secretions and excretions of water buffaloes at the time of delivery and in the following stages of lactation reported the same negative results.

## 4. Discussion

The study investigated in water buffaloes the long-term effects on the immune response following the vaccination protocol for *B. abortus* strain RB51 vaccine indicated by the WOAH Manual for the bovine species [23], modified with the use of a triple dose as previously described [26,28]. The protocol was applied to buffaloes in the prepuberal age. In particular, the present paper analyses the evolution of the humoral response using the homologous RB51 antigen for RB51-CFT, while the official RBT and CFT tests were negatives for all the duration of the trial.

The research examined the long-term effects of *B. abortus* RB51 live attenuated vaccine, administered to water buffaloes at calfhood age with a booster vaccination received at the age of 12–14 months. The few studies carried out on the use of the RB51 vaccine in water buffaloes show how the age of administration, the time interval between the first vaccine, a possible booster, and the vaccine dose used, may result in different immune response trends. This makes it is difficult to compare the studies with each other [25,33]. This is therefore important to underline that the data obtained in the present study must be interpreted and correlated with the vaccination scheme described. 

We focused on the kinetics of humoral response and assessed vaccine safety over a long-term covering from the age of prepubescent animals to the stage of adults, including pregnancy, calving, and lactation. To the best of our knowledge, data covering this whole period are not available in the international peer-reviewed literature.

As expected, both the first and the second vaccination induced seroconversion in 100% of vaccinated animals, followed by a progressive decrease in the antibody titers that, however, after the booster, never turned into complete seronegative results. In fact, over 2.5 years after the vaccine recall injection, a fluctuating serological pattern of some animals was still observed, alternating seronegative and seropositive periods, presenting antibody titers close to the threshold value. No similar observation has been reported before. Considering the triple vaccine dose administered, fluctuation of low antibodies titers may be attributed to persistence and active replication of live vaccine strain in vaccinated animals. However, bacteriological investigations, carried out using culture and PCR in parallel, excluded the presence of live vaccine strain from any of the matrix tested. Even when taking into account the low sensitivity of direct tests employed and the intermittent release the *Brucella* bacteria in milk, the long-term monitoring as well as the frequent sampling schedule applied minimized the chance of false negative test results. A more likely explanation of this fluctuation is that after declining, serum levels of RB51 antibodies remained at low concentrations, and close to the RB51-CFT threshold. Minimal fluctuations of antibody concentrations, usually within the range of one twofold dilution, may explain the variation in negative and positive results of sequential sampling. These physiological fluctuations have been reported in cattle and humans [34,35,36].

Pregnancy, calving, and lactation did not produce effects on antibody dynamics. The study also demonstrated complete safety of the vaccination schedule employed, with no seroconversion recorded in control animals, which were mixed with vaccinated animals from the first day of vaccination. No strain persistence was detected from direct investigations carried out on several biological matrices from living animals and carcasses collected during the trial (deceased animals) and at the end of the trial.

The application of the RB51 vaccination in buffalo farms of the endemic area of Caserta has been previously implemented to reduce disease prevalence before entering the eradication phase based on test and slaughter policy. This offered the opportunity to induce population immunity while maintaining the possibility to identify infected. Actually, as previously stated, RB51 vaccination does not interfere with classical serological tests (RBT and CFT) used for detection of infected animals, both for cattle and water buffaloes [30]. Our results confirm that the antibody response elicited by first and booster vaccination did not affect disease diagnosis based on official RBT or CFT, despite the triple dosage applied.

Data on kinetics of antibody response also imposed a consideration on RB51-CFT result interpretation and test application. As mentioned above, a booster vaccination, administered at the age of 12–14 months and using a triple dose, elicited a prolonged antibody response that was detected in some vaccinated animals up to 600 days’ post-booster vaccination. 

In this situation, the use in parallel of diagnostic tests that evaluate the cell mediated immunity would help in identifying vaccinated animals, as already described [29,30].

Furthermore, the kinetics of the RB51 antibody response recorded suggests that the interpretation of RB51-CFT results necessarily requires knowledge on the “official” vaccination status of the population to be investigated. In fact, the test cut-off of 1:4 was defined based on data of non-vaccinated cattle populations to maximize the specificity of the test, and after being applied to water buffaloes [8,30,37]. The results of the present study demonstrate that a number of water buffaloes receiving a booster vaccination with a triple dose at the age of 1 year or above are likely to have long-term positivity to RB51-CFT, even if in a low number. Differently, a first dose administered at 4–6 months of age is likely to induce the presence of RB51-CFT antibodies that disappear in a short time and do not persist further. This also suggests that when the test is applied to a population vaccinated with the scheduling used in the present study, antibody titers up to 1:8 in RB51-CFT are expected in some vaccinated animals, and this should be interpreted as physiological. At the same time, the individual variability observed in antibody titers suggests that RB51-CFT results have to be interpreted at the herd level and not considering individual animals only.

Another aim of the study was to assess the safety of the vaccination schedule applied, for the animals and for the environment. The results demonstrated that following vaccination, none of the control animals seroconverted and vaccine strains were never detected by tissue collected from animals deceased during the trial or slaughtered at the end to the study. Two control males were also involved in natural breeding with vaccinated female water buffaloes and negative results to both direct and indirect tests demonstrated that sexual transmission did not occur. Taking into consideration that parturition and lactation are two critical periods for disease spreading, we also monitored potential vaccine shedding during these phases. All samples (cotyledons, colostrum, and vaginal swabs) investigated during calving and lactation by culture and PCR tested negative.

A lack of RB51 isolation or direct detection throughout the study, in addition to the negative serological results of control animals left in contact with vaccinated animals, demonstrates that neither the vaccine strain was released in the environment nor a transmission between vaccinated animals and controls occurred, either through the environment or through mating.

In this study, a limited number of control animals was included. This is because the primary objective of the study was to characterize the kinetics of immune responses of water buffaloes vaccinated using a specific vaccination schedule, and a higher number of animals were allocated in the vaccinated group. In parallel, our starting hypothesis considered that the RB51 vaccine was not spread by animals vaccinated at a prepubescent age and the number of controls included, together with the stringent sampling protocol applied to both vaccinated and control animals, would have been sufficient to detect directly or indirectly any vaccine shedding. Out of the scope of the present study was the evaluation of vaccine efficacy to challenge.

## 5. Conclusions

Our study confirmed that the RB51 vaccination schedule based on the use of a triple dose administered to young animals is safe and not associated with any vaccine shedding or side effects in the long term, including pregnancy, calving, and lactation. Furthermore, we did not observe any interference with official serological tests (RBT and CFT) employed to detect *Brucella* antibodies in the context of the Italian National Brucellosis eradication plan. These data sustain the possible integration of the current test and slaughter policy with the RB51 vaccination, in high prevalence areas. Data on RB51 antibody dynamics induced by the vaccine schedule applied represent the baseline to develop diagnostic protocols aimed at monitoring the correct application of RB51 vaccination.

## Figures and Tables

**Figure 1 microorganisms-11-02078-f001:**
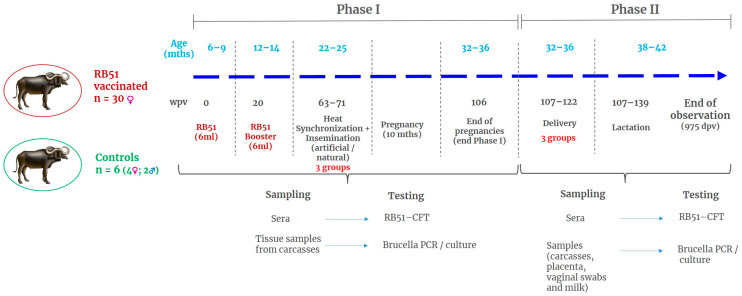
Schematic representation of the experimental design. Information on different steps, timing, age of the animals, and physiological stage are displayed.

**Figure 2 microorganisms-11-02078-f002:**
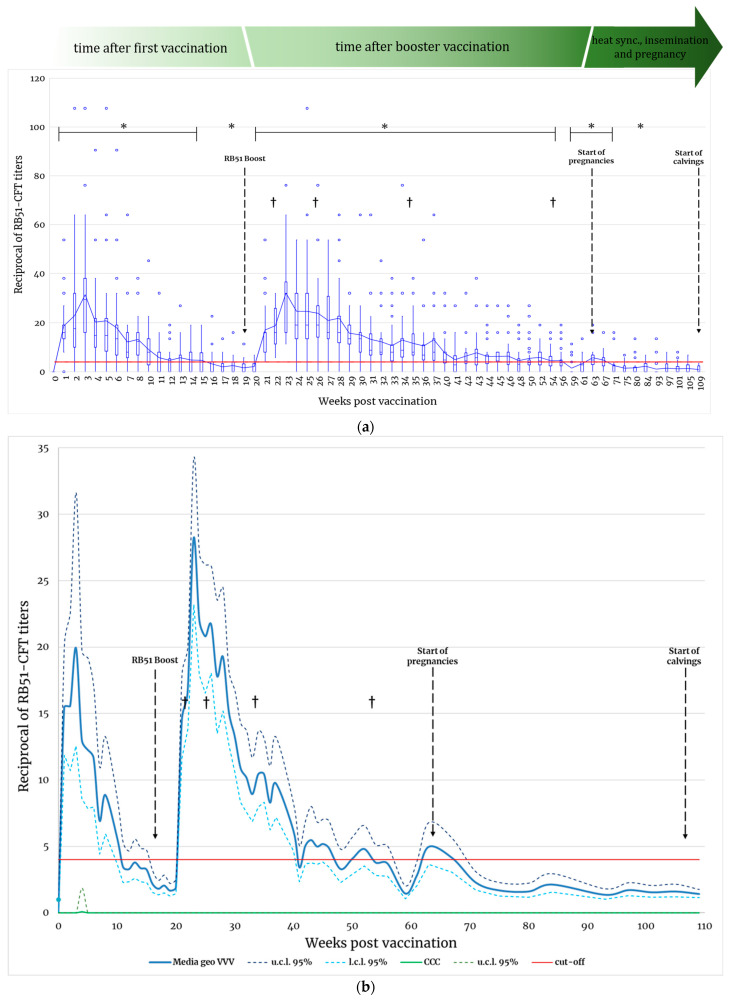
(**a**) Antibody response to RB51 vaccination measured by RB51-CFT from vaccination to calving. Values are expressed as the reciprocal of serum dilutions. The box and whisker-plot chart shows distribution RB51-CFT data into quartiles. Lines inside the boxes indicate group median value. Outliers are displayed as blue circles. The blue line indicates the mean of grouped values from vaccinated animals. The red line shows the test cut-off (1:4). Crosses represent animal death. Non-parametric Mann–Whitney test was applied to compare Ab titers between vaccinated and control groups and significant differences (*p* < 0.05) in are indicated with “*”. Data from control group are not displayed and remained constantly below the cut-off value. (**b**) Geometric mean of antibody response to RB51 vaccination measured by RB51-CFT from vaccination to calving. Values are expressed as reciprocal of serum dilutions. The continuous blue line indicates the mean of grouped values from vaccinated animals (VVV). Dotted dark blue and azul lines represents upper (u.c.l.) and lower (l.c.l.) confidential limits respectively of vaccinated animals. The continuous green line indicates the mean of grouped values from control animals (CCC). The red line shows the test cut-off (1:4). Crosses represent animal death.

**Figure 3 microorganisms-11-02078-f003:**
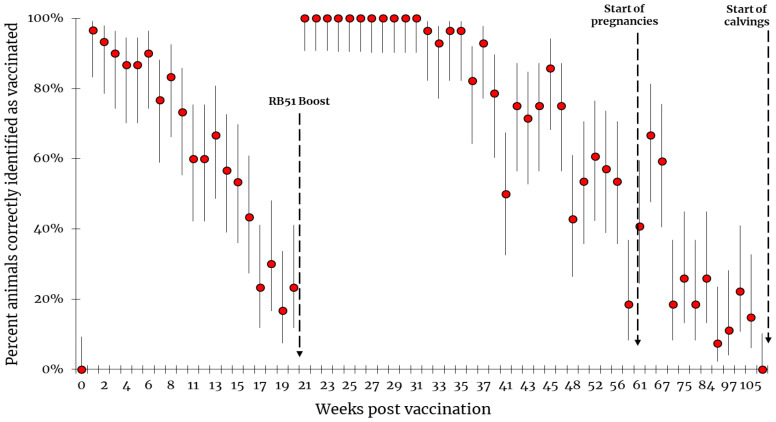
Variations in RB51-CFT sensitivity after vaccination, in terms of percentage of vaccinated animals correctly identified as vaccinated. Data refer to the period from vaccination to calving. Results are expressed as the percentage of vaccinated animals resulting positive to RB51-CFT (Ab titer ≥ 1:4) over the number of vaccinated animals tested for each time point. Bars indicate upper and lower confidential limits calculated at 95% confidence interval using a Bayesian approach with a beta distribution.

**Figure 4 microorganisms-11-02078-f004:**
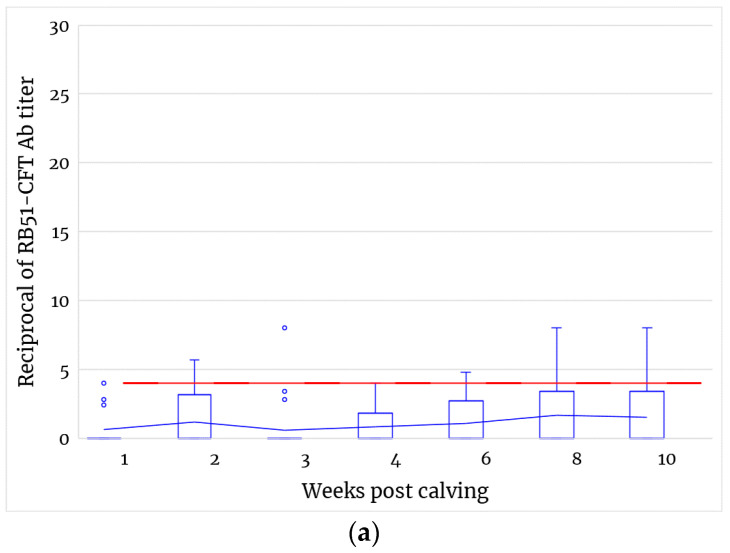
(**a**) RB51-CFT antibody titers observed after calving and during lactation. Values are expressed as reciprocal of serum dilutions. The box and whisker chart shows distribution RB51-CFT data into quartiles. Lines inside the boxes indicate group median value. Outliers are displayed as blue circles. The blue line indicates the mean of grouped values from vaccinated animals. The red line shows the test cut-off (1:4). Non-parametric Mann–Whitney test was applied to compare Ab titers between vaccinated and control groups. No significant differences were observed between vaccinated and control animals. Data from control group are not displayed and remained constantly below the cut-off value. (**b**) Percentage of animals testing positive to RB51-CFT after calving and during lactation. Results are expressed as the percentage of vaccinated animals resulting positive to RB51-CFT (Ab titer ≥ 1:4) over the number of vaccinated animals tested for each time point. Bars indicate upper and lower confidential limits calculated at 95% confidence interval using a Bayesian approach with a beta distribution.

**Table 1 microorganisms-11-02078-t001:** Number of samples negative to *Brucella* spp. tested by microbiological isolation and PCR in different stages of the trial.

Animals (n)	Stage of the Trial	Matrices	Samples (n)
4	Post vaccination (dead animals)	lymph nodes	31
		spleen	4
		udder	3
		other organs	14
29	Calving	cotiledons	28
		colostrum	27
		vaginal swabs	29
29	Lactation	milk	733
		vaginal swabs	369
29	End of trial (slaughtered animals)	lymph nodes	92
		spleen	22
		udder	21
		other organs	5

## Data Availability

Data are contained within the article.

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
