# Peer review of "Brucella abortus Strain RB51 Administered to Prepubescent Water Buffaloes, from Vaccination to Lactation: Kinetics of Antibody Response and Vaccine Safety"

_microorganisms, 2023, doi:10.3390/microorganisms11082078_

Round 1
Reviewer 1 Report
The present study focused on kinetics of humoral response and also assessed vaccine safety during pregnancy, calving and lactation.
The results of the present study confirm that after RB51 vaccination none of the vaccinated animals reacted positive to classical RBT or CFT performed according to the WOAH Manual.
The study provided valuable findings concerning brucellosis which is a serious infection of zoonotic and economic concerns. RB51 is a commonly used vaccine in bovine with adequate available information in cattle, while information is lacking in buffaloes. However, this study has some points that need revisions by authors.
- Reason of using triple dose vaccination is needed?
- Lines 23-24 “for more months……” what is this mean? Time should be specified accurately.
- Sentence at line 70, this is repeated information in lines 56 - 57 of introduction.
- In introduction, the authors focused on mainly on the disadvantages and harmful effects of RB51 vaccine. Alternatively, authors should focus on the mechanism of immunoprotection of RB51 vaccine.
- In materials and methods, detailed method of blood sample collection, serum preparation, collection and storage and used serum dilution should be described.
- In discussion, why animals converted to seronegativity after first vaccination and testing seropositive?
- Conclusion should be summarized.
The study is written well but still need minor English editing and revisions.
Reviewer 2 Report
In this work, the Brucella RB51 vaccine strain in buffaloes was used to develop a vaccination protocol for this species. The kinetics of the antibody response and the potential excretion of the bacterium were evaluated in the period following vaccination. Additionally, the authors highlight a safe vaccination schedule that allows discrimination between vaccinated and unvaccinated animals. The article is well written and easy to understand for a wide audience
Some points should be reviewed to improve the article and make it more compressible:
- The introduction misses the importance of dairy products in the local and global economy as well as the relevance of the water buffalo has in human health.
- In section 2.1 was not specify the age of the buffaloes used in the experiment.
- In figure 2a should be useful to integrate the information reported in figure 1 concerned the representation of the experimental design, in order to better follow the evolution of antibody during different phases
- In section 2.4.2 was not specify which type of antibody targeting the serological test (IgG, IgA, IgM).
- In line 228 the s word missing after “tested samples and”.
- A review of the use of the terms “Brucella” and “brucellosis” is required. Brucella is written not in italics (lines 294, 343) and brucellosis is written with a capital letter (line 193).
